# Self-Reported Social Relationship Capacities Predict Motor, Functional and Cognitive Decline in Huntington’s Disease

**DOI:** 10.3390/jpm12020174

**Published:** 2022-01-27

**Authors:** Pablo Lemercier, Laurent Cleret de Langavant, Jennifer Hamet Bagnou, Katia Youssov, Laurie Lemoine, Etienne Audureau, Renaud Massart, Anne-Catherine Bachoud-Lévi

**Affiliations:** 1Département d’Etudes Cognitives, Ecole Normale Supérieure, PSL University, 75005 Paris, France; pablo.lemercier@inserm.fr (P.L.); laurent.cleret@gmail.com (L.C.d.L.); hametj.npi@gmail.com (J.H.B.); katia.youssov@aphp.fr (K.Y.); laurie.lemoine@aphp.fr (L.L.); rmassart.npi@gmail.com (R.M.); 2Equipe NeuroPsychologie Interventionnelle, Institut Mondor de Recherche Biomédicale, INSERM U955, University Paris Est Créteil, 94000 Créteil, France; 3Centre National de Référence Maladie de Huntington, Service de Neurologie, Hôpital Henri Mondor-Albert Chenevier, AP-HP, 94000 Créteil, France; 4NeurATRIS, 94000 Créteil, France; 5Clinical Epidemiology and Ageing, Service de Santé Publique, Henri Mondor Hospital, AP-HP, 94000 Créteil, France; etienne.audureau@aphp.fr

**Keywords:** Huntington’s disease, social relationships, disease progression, auto-questionnaire

## Abstract

Huntington’s Disease (HD) is an inherited neurodegenerative disease characterized by a combination of motor, cognitive, and behavioral disorders. The social and behavioral symptoms observed in HD patients impact their quality of life and probably explain their relational difficulties, conflicts, and social withdrawal. In this study, we described the development of the Social Relationship Self-Questionnaire (SRSQ), a self-reporting questionnaire that assesses how HD patients perceived their social relationships. The scale was proposed for 66 HD patients at an early stage of the disease, 32 PreHD patients (individuals carrying the mutant gene without motor symptoms), and 66 controls. The HD patients were included in a prospective longitudinal follow-up for an average of 1.07 years with motor, functional, cognitive, and behavioral assessments. Based on the HD patients’ answers at baseline, we identified two domains in the SRSQ. The first domain was related to social motivation and correlated with cognitive performance. The second domain was related to emotional insight and correlated with behavioral symptoms such as apathy, anxiety, and irritability. We discovered that both SRSQ domain scores at baseline predicted future motor, functional, and cognitive decline in HD.

## 1. Introduction

Huntington’s Disease (HD) is an inherited neurodegenerative disease caused by the expansion of CAG repeats in the huntingtin gene. HD typically becomes symptomatic in mid-life and associates a combination of motor (chorea, gait disorder), cognitive (executive dysfunction, memory, and language disorders), and behavioral (depression, apathy, irritability, delusions, aggressivity, obsessions-compulsions, and anosognosia) disorders, which lead to severe disability.

A wide spectrum of social and behavioral symptoms is observed in HD, ranging from mild to severe. Behavioral disorders are frequent in HD with a prevalence between 33% and 76% [1]. These disorders may occur at different stages of the disease. Depression and irritability often occur early in the disease, sometimes long before the onset of motor or cognitive dysfunction, but typically do not worsen with disease evolution [2]. Conversely, apathy, which can be present at the earliest stages of the disease, often worsens and becomes more prevalent later on [1,3]. HD patients also show impairments in social cognition [4], i.e., socio-emotional capacities and experiences that regulate relationships between individuals [5]. HD patients commonly show deficits in emotion processing, but also in theory of mind, which is the ability to infer the mental states of others, while empathy could be less affected [6].

These social and behavioral symptoms probably explain HD patients’ relational difficulties, conflicts, and social withdrawal. These symptoms may have a greater impact on the quality of life of patients and their families than motor symptoms [2,7]. Accordingly, families of HD gene carriers identify social and behavioral symptoms as the most debilitating disorders even during the presymptomatic stage (PreHD) [8,9]. A meta-analysis confirmed that the deficit in emotion recognition in HD, a social disorder, is associated with a high disease burden but also likely with the early onset of motor symptoms, and cognitive impairments [4]. Another recent study found that being alone and not accompanied by a relative to a clinical visit was the main predictor of HD progression, further suggesting the prognostic importance of the social network in this disease [10]. As a whole, social and behavioral disorders are important for patients’ quality of life and could be used as prognostic biomarkers in HD [4,11].

In this study, we developed and validated an auto-questionnaire to assess HD patients’ own perceptions of their social relationships, the Social Relationship Self-Questionnaire (SRSQ). The uniqueness of this approach relies on the investigation of both positive and negative questions related to social relationships involving the patient’s perspective, rather than symptoms reported by a professional. We showed that the SRSQ measures HD patients’ perceived quality of their social relationships, and may also serve to predict future cognitive, motor, and functional decline in HD.

## 2. Materials and Methods

### 2.1. Auto-Questionnaire Development

To build the Social Relationship Self-Questionnaire (SRSQ), we followed steps designed to ensure clinical and psychometric validity. In November 2013, we first set up a focus group composed of health professionals who specialized in HD patients’ follow-up (a neurologist, a neuropsychologist, and a clinical psychologist), family relatives, and representatives of HD patients’ lay associations. Before the first meeting, each participant sent a set of the twenty most important issues or questions that, in their view, related to patients’ social relationships or to their relationships with patients. In addition, ten patients with HD were interviewed separately by a clinical psychologist to integrate their points of view in the future questionnaire. The different perspectives of HD patients and people familiar with HD were summarized and presented to the focus group. The proposals were compiled and grouped into domains to structure the discussion and elicit questions related to each important issue (empathy, altruism, confidence, attention to others, listening to others, withdrawal, adaptability, coping with illness/lost hope, nervosity, irritability, joy, pleasure, social engagement, perspective-taking capacity, self-confidence, sensitivity to other’s gazes and judgment, self-awareness, isolation, and miscellaneous). Group discussions were guided by two semi-structured research questions: (1) According to your experience, are these categories relevant for a good quality of social relationships? (2) Are these items well formulated and easy to understand? The participants proposed 122 sentences illustrating their views on the different issues.

The sentences were discussed by clinicians and researchers from the laboratory of *NeuroPsychologie Interventionnelle* (Créteil, France) in order to check if they captured the most important issues while avoiding redundancy, offense, and ambiguity. Two experts in social cognition provided a final rephrasing to balance between positive and negative sentences to prevent response biases. For some sentences, an answer at the extreme of the scale represents social awareness and positive feelings about social relationships (i.e., “I am satisfied with the help provided by my next of kin”). Other sentences are written in such a way that a response at the same extreme represents social awareness and negative feelings about social relationships (for example, “My relatives are not doing enough to help me”). A total of 49 sentences was obtained (see Appendix A—Selected Items of the SRSQ).

During the first half of 2014, an online pilot survey was conducted to assess the quality of the selected sentences in 77 healthy adult controls (37 males and 40 females; mean age: 32.1; SD: 9.7) and 17 HD mutation carriers (12 HD and 5 PreHD: 12 males and 5 females; mean age: 45.3. SD: 9.6). Eligible individuals were able to read and write in French and had sufficient cognitive ability to complete the questionnaire. Participants had to indicate how much each sentence represents what he or she feels. A six-point Likert scale was used (“absolutely true”, “true”, “mostly true”, “mostly false”, “false”, “absolutely false”). To preserve the intimacy of the subjects, two items, “I am satisfied with my sex life” and “My partner is not satisfied with his or her sex life”, were optional. After completing the SRSQ, the participants were interviewed and the following points were addressed: their general impressions, the completeness of the questionnaire, the clarity of the instructions, the choice of possible answers, the readability of the format, and the participants’ interpretation of each item. They were also asked if the main aspects of social relationships in HD patients were explored in the questionnaire.

This pilot resulted in a final version of the questionnaire, in which ambiguous or unnecessary items were removed, while other items were added or rephrased to better explore an important issue.

### 2.2. Validation Study

A validation study was conducted on a prospective longitudinal cohort to check the psychometric properties of the SRSQ.

#### 2.2.1. Participants

For the development and the validation of the SRSQ, participants were recruited from several studies: a validation study specific to this questionnaire, the global Enroll-HD study (http://www.enroll-hd.org, accessed on 10 December 2021), the Predictive Biomarkers for Huntington’s disease study (BIOHD; NCT01412125) and the CAPIT-HD2 study (NCT03119246). Ethical approval for all studies was granted by the local ethics committee of *Comité de Protection des Personnes* (CPP) *Ile-de-France III* (Enroll-HD: study number: 3188, IDRCB: 2014-A01276-41, 2018/11/06; BIOHD: study number: P090302, CPP reference: Aam8754-5-Mondor03-02, 2021/01/08; CAPIT-HD2: study number: P150201, EUDRACT: 2016-A00308-43, CPP reference: Am8686-5-3376, 2020/10/13). The participants were provided oral and written information about the research and gave their written informed consent before their participation and after the study had been fully explained to them, in compliance with research standards for human research and in accordance with the Helsinki Declaration. All patients were recruited at Henri Mondor hospital (Créteil, France).

The inclusion criteria for patients were (1) genetically confirmed HD (≥ 36 CAG repeats), (2) completion of the Unified Huntington’s Disease Rating Scale (UHDRS) within 90 days before or after the first survey completion, and (3) having UHDRS Total Functional Capacity (TFC) scores (TFC ≥ 7) at inclusion. The PreHD stage was acknowledged for patients with a TFC = 13 and a UHDRS Total Motor Score (TMS) < 5 [12]. Healthy controls were recruited from the general population. We excluded subjects who lacked information on age, sex, and years of education. Additional exclusion criteria included having other neurological diseases. The participants from the control group were free of psychiatric disorders and did not show uncontrolled disability. All participants completed a paper version of the SRSQ. All participants were able to answer the questionnaire without assistance.

Between September 2014 and February 2021, a total of 219 subjects (152 patients and 67 controls) answered the SRSQ. The final analyses were conducted on 66 HD, 32 PreHD, and 66 controls that met the selection criteria. A general description of the cohort is provided in Table 1.

#### 2.2.2. Scale Presentation

The original SRSQ scale was an auto-questionnaire comprised of a set of 49 items with a six-point Likert scale for responses (from −2.5 to 2.5: absolutely false, false, mostly false, mostly true, true, and absolutely true). Instructions were: “Each sentence describes a situation. To indicate your level of agreement with a statement, check the box that best describes how you feel. Try to be as accurate as possible. Ignore what others might think or want, focus only on the way you feel today. Your answers will be kept confidential”.

### 2.3. Clinical Measures

Clinical severity of HD was assessed according to the UHDRS [13]. The Total Motor Score (TMS) from this assessment scored higher in cases of motor symptoms (max: 124). Daily function was assessed by Total Functional Capacity scores (TFC; low scores indicated poorer function, max: 13), Functional Assessment scores (low scores indicated poorer function, max: 50), and the Independence Scale (low scores indicated poorer function; range 0–100). Cognitive assessment included the Literal and Categorical Verbal Fluency Task, the Symbol Digit Modalities Test (SDMT), the three-part Stroop Tests (color, word, and interference) from the UHDRS. Global cognition was assessed in accordance with the Mattis Dementia-Rating Scale (MDRS) [14], and verbal episodic memory was assessed in accordance with the Hopkins Verbal Learning Memory Test B [15].

The overall burden of the disease was assessed using the CAG age product (CAP) score [16]:CAP score = age × (CAG repeats−33.66)(1)

The overall disease severity was assessed with the Composite Unified Huntington’s Disease Rating Scale (cUHDRS) composite score as follows [17]:(2)$$\mathrm{cUHDRS}=\left[\left(\frac{\mathrm{TFC}-10.4}{1.9}\right)-\left(\frac{\mathrm{TMS}-29.7}{14.9}\right)+\left(\frac{\mathrm{SDMT}-28.4}{11.3}\right)+\left(\frac{\mathrm{Stroop}\;\mathrm{Word}-66.1}{20.1}\right)\right]+10$$

The most common behavioral problems in HD were determined via semi-structured interviews of patients and their family participants by a clinician using the 11-item short version of the Problem Behaviors Assessment Scale (PBA-s) [18]. The behavioral severity scores were used (higher scores indicating poorer outcome; max: 5). In addition, the self-administered Hospital Anxiety and Depression Scale (HADS) was used [19].

### 2.4. Statistical Analysis

#### 2.4.1. Exploratory Factor Analysis

Psychometric analysis was performed on HD participants’ responses to the SRSQ and included assessment of item characteristics, construct validity, internal consistency, known-groups validity, and convergent validity. This analysis was carried out on the original SRSQ excluding the two optional items, leaving a total of 47 items.

Descriptive analyses were performed to study the distribution of individual items, to assess acceptability (% missing values), and to identify potential ceiling and/or floor effects when a majority of item responses were distributed at either end of the scale. Then, items with missing data were imputed using the k-Nearest Neighbor algorithm applied to the 47 investigated items while adjusting for sex, age, and education.

The results of a Bartlett’s test of sphericity (*p*-value < 0.001) and a Kaiser–Meyer–Olkin measure of sampling adequacy (0.41) indicated that an exploratory factor analysis (EFA), a principal factor method with non-orthogonal oblique rotation also known as oblimin rotation, could be conducted to examine the underlying constructs of the scale and to characterize its dimensionality [20]. First, to determine the optimal number of sets of items (also called domains or factors) making up the scale, we performed an Horn’s parallel analysis [21] based on the 95th percentile estimate and computed a Velicer’s minimum average partial (MAP) criterion [22]. These metrics suggested that items should be grouped into two domains when performing the EFA. Items were considered for deletion if their factor loadings were <0.4 for each domain, and/or if their communalities were <0.3 (uniqueness > 0.7).

Internal consistency reliability (homogeneity of the items) was assessed by calculating Cronbach’s alpha. A coefficient score of above 0.8 indicated good internal consistency and a score above 0.9 indicated excellent consistency.

#### 2.4.2. Description of Groups

The normality of the descriptive variables was checked visually. To describe the PreHD, HD, and control groups at baseline, we reported the mean and the standard deviation (SD) for continuous variables, whereas we reported sample size and percent for categorical variables (Table 1). Pairwise differences between groups were tested with Student’s and Chi-squared tests.

Longitudinal changes over time were estimated in HD with linear mixed models with a random intercept (Table 2). Age, sex, CAG number, and education were selected as adjustment variables. Thus, we computed the estimated marginal means to estimate the average annual change over the follow-up period after removing the effect of covariates.

#### 2.4.3. Groups Comparison on SRSQ Scores

At baseline, we tested for statistical differences between the PreHD, HD, and control groups on SRSQ scores, with linear models adjusted on the basis of age, sex, and education. Then, we performed post hoc Student’s *t*-tests to evaluate the estimated marginal means difference between each pair of groups.

In addition, we assessed whether the three groups had significant changes in SRSQ scores over the follow-up period. We used linear mixed models with a random intercept adjusted according to age, sex, and education. The effect of time and the interaction between group and time (Group*Time) were tested to investigate whether the scores’ trajectories varied over time and between groups, respectively.

#### 2.4.4. Correlation between SRSQ Scores and Motor, Functional, and Cognitive Abilities

To investigate the relationship between SRSQ scores and motor, functional, and cognitive abilities, we ran Pearson’s correlations in the HD group and the PreHD group at baseline.

We further assessed the relationship between SRSQ scores and motor, functional, and cognitive repeated measurements that were acquired during the follow-up, using linear mixed models [23,24] with random intercept adjusted according to age, sex, CAG, and education.

#### 2.4.5. Prediction of the Disease Progression

Finally, we sought to explore whether SRSQ scores could predict longitudinal changes of motor, functional, and cognitive abilities in HD participants only (Table 1). We used linear mixed models with random intercepts adjusted according to age, sex, CAG, and education. We tested for the statistical significance of the coefficients of interaction between the domain scores measured at baseline and time, to determine whether SRSQ can predict clinical trajectories. The effect sizes were estimated using partial Cohen’s f^2^, which represented the amount of variance of the response variables (the outcome) that is explained by the explanatory variables (the SRSQ), after accounting for other predictors in the model [25].

The normal distribution of residuals, random effects, and the homoscedasticity of residuals were reviewed for each model. Degrees of freedom of linear mixed models were determined according to the Satterthwaite approximation test. All statistical analyses were performed using R software, version 4.0.5, including the “VIM”, “psych”, “lme4”, “lmerTest”, “emmeans”, and “effectsize” packages, all of which are available at http://cran.r-project.org/web/packages (accessed on 10 December 2021).

## 3. Results

### 3.1. Exploratory Factor Analysis

The exploratory factor analysis conducted for HD participants identified two domains of 12 items each (Figure 1; Table 3). The first domain related to social motivation and described a subject’s willingness to connect with others, wherein high values are associated with high social motivation. The second domain related to emotional insight and described a subject’s feelings regarding social experiences with others, with high values denoting negative emotional insight.

We calculated a score for the first and the second domains. Items negatively associated with their domain had their responses score reversed before all items were summed. Thereafter, they were referred to as the social motivation score and the emotional insight score, respectively. These scores are both bounded between −30 and 30.

Cronbach’s alpha showed good internal consistency on the 24 selected items (alpha = 0.87; 95% CI = (0.83–0.92)), the first domain (alpha = 0.84; 95% CI: 95% = (0.79–0.90)), and the second domain (alpha = 0.87; 95% CI = (0.82–0.91)).

### 3.2. Relationship between SRSQ and Clinical Scores

Pearson’s correlations showed that the two SRSQ scores are negatively correlated with each other (*r* = −0.30, 95% CI = (−0.50; −0.06), *p*-value = 0.015). In addition, we observed moderate but significant associations between the SRSQ social motivation score and cognitive scores and the cUHDRS composite score in both HD and PreHD groups. We found significant associations between the SRSQ emotional insight score and HADS Irritability, HADS Anxiety, and PBA apathy scores in HD patients only. See the Table A1 (HD) and Table A2 (PreHD) provided in Appendix B. Examples of the cross-sectional relationships observed in HD and PreHD are illustrated in Figure 2.

Follow-up duration, number of subjects, and number of visits by tasks are described in Table 2. We tested whether the SRSQ scores in HD patients are related to motor, functional, and cognitive abilities by measuring associations in repeated measures. Detailed follow-up results are provided in Table A3 from Appendix B. We showed that the social motivation score correlates with literal verbal fluency, Hopkins B memory performance, the MDRS total score, and the HADS scores (depression and anxiety). The emotional insight score correlates with HADS (depression, irritability, and anxiety) and PBA irritability scores (see Table A3 from Appendix B).

### 3.3. SRSQ Scores between Groups

At baseline, we assessed whether SRSQ scores differed between HD, PreHD, and control groups. We did not observe any significant differences between groups for the social motivation score (HD versus control: *p*-value = 0.057; HD versus PreHD: *p*-value = 0.267; and PreHD versus control: *p*-value = 0.663) or the emotional insight score (HD versus control: *p*-value = 0.821; HD versus PreHD: *p*-value = 0.780; and PreHD versus control: *p*-value = 0.921).

At follow-up, the social motivation domain score did not change over time for any group (HD: *p*-value = 0.235; PreHD: *p*-value = 0.944; and control: *p*-value = 0.379). Likewise, the emotional insight score did not change over time for any group (HD: *p*-value = 0.232; PreHD: *p*-value = 0.397; and control: *p*-value = 0.213). These results suggest that the SRSQ scores are stable over an average follow-up of one year.

### 3.4. Prediction of the Disease Progression

Linear mixed models showed significant interactions between the social motivation score and time on motor, functional, and cognitive changes over time in HD patients (Table 4; Figure 3). Likewise, we observed significant interactions between the emotional insight score and time on functional and cognitive changes over time. Overall, these results suggest that participants with high scores on social motivation and/or low scores on emotional insight at baseline have higher changes over time (rapid disease progression). Cohen’s f^2^ values revealed small- to medium-effect sizes for the interactions between the social motivation score and time, and small-effect sizes for the interactions between the emotional insight score and time.

## 4. Discussion

### 4.1. Summary of the Results

In this study, we described the development of the SRSQ, a self-reporting questionnaire assessing social relationships in HD patients. Based on the answers from a prospective longitudinal cohort of 66 HD patients at early stages, we identified two domains. The first domain was related to social motivation and correlated with executive function and global cognitive performance. The second domain was related to emotional insight and correlated with behavioral symptoms such as apathy, anxiety, and irritability. The overall scores did not differ between the three groups (control, PreHD, and HD). Longitudinal assessment of SRSQ did not show significant change over time in any group. However, SRSQ domain scores at baseline predicted future motor, functional, and cognitive decline in HD patients.

### 4.2. The SRSQ Meaning

The SRSQ assessed for the first time HD patients’ own perceptions about their social relationships, with two consistent domains, social motivation and emotional insight. The labels chosen for the two domains were consistent with the clustering of items from factorial analysis and with the HD patients’ most frequent responses. For example, the social motivation domain included the item “I like to make my relatives smile and laugh”, which was true for 57/66 HD patients (86.4%) in our sample. The social motivation domain described the participant’s willingness to engage in social relationships. The emotional insight domain included the item “My family thinks that I am deliberately irritable or angry”, which was false for 55/66 HD patients (83.3%). The emotional insight domain described the participant’s feelings about social relationships. Interestingly, these domains correlated differently with clinical assessment variables. The social motivation domain mostly correlated with executive function and global cognition, but also with cUHDRS, which includes cognitive measures, and with depression, a condition associated with executive dysfunction. Conversely, the emotional insight domain mostly correlated with anxiety, irritability, and apathy measures.

The SRSQ is a subjective questionnaire and the HD patients’ responses should be considered with caution. For example, the SRSQ could be affected by denial, lack of awareness, or anosognosia often present in HD patients [26]. For these reasons, the SRSQ cannot be readily used as an objective estimation of social cognition performance in HD patients.

Importantly, both SRSQ domains remained stable over one year during our study, while motor, cognitive, functional, and composite assessment measures showed a significant decline over the same period. Therefore, SRSQ scores cannot be used as progression biomarkers. Moreover, the overall scores for the three groups (control, PreHD and HD) are similar, so the SRSQ scores were not disease biomarkers and probably cannot be used to detect the clinical onset of HD. However, we discovered that baseline SRSQ scores can predict future outcomes in HD patients.

### 4.3. Utility of SRSQ for Longitudinal Follow-Up

HD patients with the highest scores in the social motivation domain and/or the lowest scores in the emotional insight domain at baseline showed steeper cognitive, motor, functional and composite decline at follow-up. Although this result could be at odds with clinical expectations, it could be interpreted using the concept of cognitive reserve that was developed in the context of dementia [27,28]. The concept of cognitive reserve explains why some individuals with severe neuropathological lesions of Alzheimer’s disease in their brain show no apparent symptoms of dementia. Cognitive reserve might reflect the quality of neuronal connectivity and correlate with education and curiosity skills. Individuals with high cognitive reserve better resist brain lesions and develop disease symptoms later in life, compared to individuals with low cognitive reserve. A protective effect of cognitive reserve has also been reported in HD patients [29]. However, once the disease symptoms appear in people with high cognitive reserve, the decline is steeper than in people with low cognitive reserve. We suggest that high SRSQ scores could reflect a form of social reserve in HD patients, with similar implications for follow-up: a high social reserve at baseline would be associated with future steeper decline. Because we did not find any correlation between SRSQ scores and education in our study, we propose that such social reserve is distinct from classical cognitive reserve. This social reserve remains a novel concept, although a few papers coined this term to account for the social network of an individual (the number of persons in contact with the individual) or for the social cognition skills that remain available in AD patients [30]. Following our proposition, we could expect PreHD with higher social reserve, according to the SRSQ, to have a later clinical onset of their disease.

### 4.4. Limitations of Our Study

Despite an important number of participants being contacted for the development of the SRSQ, the final number of patients remained relatively small. In addition, we had few participants with all of the clinical data, aa they come from different cohort studies. A replication of the utility of the SRSQ in a larger cohort and in another center would be of interest. Despite promising preliminary results in other languages (English and German), the SRSQ is currently validated in the French language and further studies are necessary to replicate its efficacy in other languages.

Future uses of the SRSQ would benefit from a comparison of the SRSQ scores with social cognition tests (theory of mind, empathy, and emotion processing), quality-of-life scales, and anosognosia assessments. Social network and objective quantification of social contacts would be of interest to better delineate the concept of social reserve.

### 4.5. Perspectives and Conclusion

Contrary to existing tools assessing the quality of life in HD patients, we created a questionnaire with both positive and negative questions related to social relationships according to the patient’s perspective [31,32]. Few questionnaires have been developed to measure the impact of neurodegenerative diseases on social functioning. They provide an objective quantification of the patient’s social interactions, both in society and in their personal environment [33]. Our questionnaire focuses instead on the subjective feelings of HD patients regarding social relationships. To our knowledge, this subjective perspective has not yet been similarly assessed in other neurodegenerative diseases.

The SRSQ is brief enough to be administered in a waiting room and provides critical information on the patient’s outcomes. The SRSQ may help clinicians identify patients at higher risk of future decline who will require particular attention and set up interventions to anticipate needs for assistance in a professional context and/or in daily life. In addition, identification of participants at higher risk of decline on the basis of behavioral symptoms may improve participant selection in clinical trials seeking to find treatments to delay the onset or slow the progression of early pathological changes.

Important future research will be able to test the possibility that high SRSQ scores in PreHD individuals is associated with later onset of clinical symptoms, due to social reserve. Another related issue would be influencing such social reserve through social stimulations to delay the clinical onset of HD. Finally, testing the SRSQ in relation to other diseases, such as Alzheimer’s disease or Parkinson’s disease, would be of interest because brain neurodegenerative diseases share many clinical similarities.

## Figures and Tables

**Figure 1 jpm-12-00174-f001:**
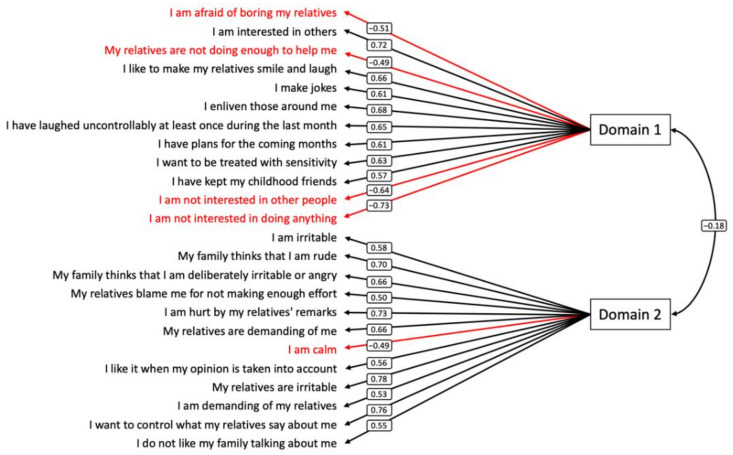
Path diagram of the final exploratory factor analysis. Black and red one-headed arrows represent positive and negative standardized loadings, respectively. The two-headed arrow represents correlation. Domain 1 reflects social motivation and Domain 2 reflects emotional insight.

**Figure 2 jpm-12-00174-f002:**
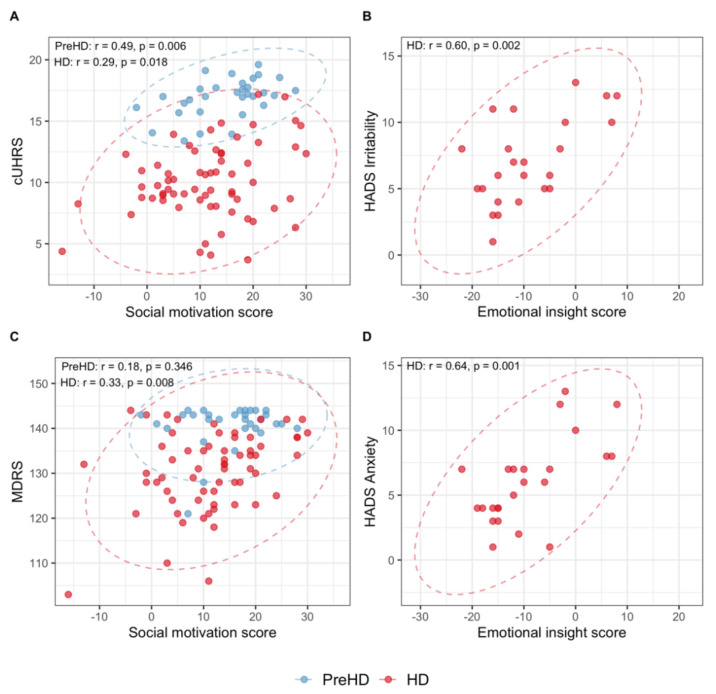
Relationships between SRSQ domains and clinical assessments: (**A**) social motivation and cUHDRS; (**B**) emotional insight and HADS Irritability; (**C**) social motivation and MDRS; (**D**) emotional insight and HADS Anxiety. Abbreviations: PreHD = presymptomatic stage, HD = Huntington’s Disease, cUHDRS = Composite Unified Huntington’s Disease Rating Scale, HADS = Hospital Anxiety and Depression Scale, MDRS = Mattis Dementia-Rating Scale.

**Figure 3 jpm-12-00174-f003:**
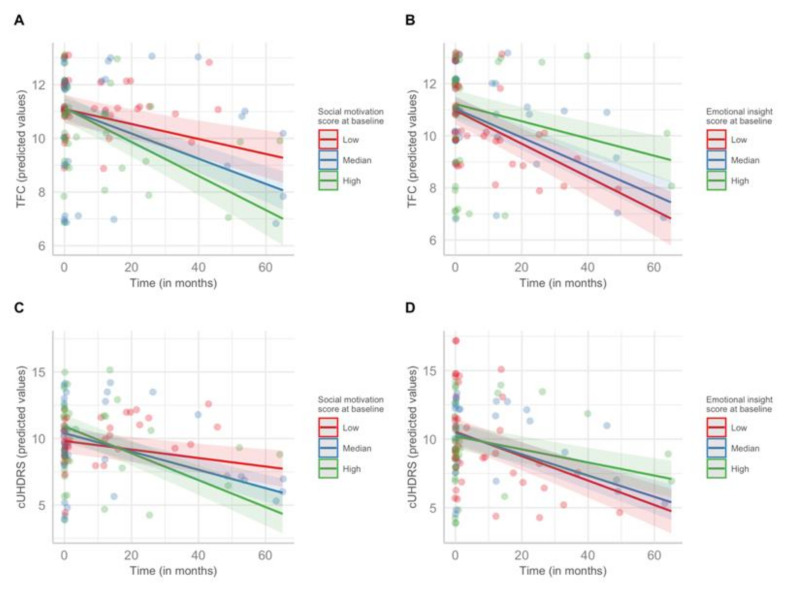
SRSQ scores at baseline and disease progression in HD patients only: (**A**) social motivation and predicted values of TFC (Total Functional Capacity) over time; (**B**) emotional insight and predicted values of TFC over time; (**C**) social motivation and predicted values of cUHDRS (Composite Unified Huntington’s Disease Rating Scale) over time; (**D**) emotional insight and predicted values of cUHDRS over time. Predicted values of TFC and cUHDRS are calculated using the estimated marginal means to adjust on the basis of age, sex, number of CAG repeats, and education level. Low, median, and high SRQS scores correspond to the first quartile, the median, and the third quartile, respectively. Colored bands represent 95% confidence interval.

**Table 1 jpm-12-00174-t001:** Description of participants.

Variable	HD *n* = 66	PreHD *n* = 32	Control *n* = 66	Pairwise Comparison ^2^		
				HD vs. PreHD	HD vs. Control	PreHD vs. Control
**Age**, mean ± SD	51.0 ± 11.5	44.9 ± 11.8	45.4 ± 12.6	0.020 *	0.008 **	0.850
**Sex**, *n* (%) of men	31 (47%)	14 (44%)	27 (41%)	0.933	0.599	0.961
**Years of education**, mean ± SD	13.3 ± 2.8	14.7 ± 2.3	13.8 ± 2.7	0.021 *	0.296	0.138
**CAG repeats**, mean ± SD	44.6 ± 4.0	42.7 ± 2.8		0.014 *		
**CAP score**, mean ± SD	525 ± 102	384 ± 88.9		<0.001 ***		
**TFC**, mean ± SD	11.0 ± 1.6	13.0 ± 0	13.0 ± 0 ^1^	<0.001 ***	<0.001 ***	1.00
**Functional Assessment**, mean ± SD	22.5 ± 3.0	25.0 ± 0	25.0 ± 0 ^1^	<0.001 ***	<0.001 ***	1.00
**Independence Scale**, mean ± SD	88.79 ± 10.2	100 ± 0	100 ± 0 ^1^	<0.001 ***	<0.001 ***	1.00
**TMS**, mean ± SD	31.0 ± 13.6	1.1 ± 1.5	0.4 ± 0.7 ^1^	<0.001 ***	<0.001 ***	0.766
**Stroop Word**, mean ± SD	65.7 ± 16.7	96.3 ± 14.2	103.2 ± 16.6 ^1^	<0.001 ***	<0.001 ***	0.961
**Stroop Color**, mean ± SD	46.9 ± 15.2	76.1 ± 12.5	78.2 ± 10.1 ^1^	<0.001 ***	<0.001 ***	0.541
**Stroop Interference**, mean ± SD	26.9 ± 10.2	46.8 ± 11.1	45.6 ± 10.3 ^1^	<0.001 ***	<0.001 ***	0.644
**SDMT**, mean ± SD	26.7 ± 10.5	52.4 ± 11.4	51.3 ± 8.2 ^1^	<0.001 ***	<0.001 ***	0.664
**Verbal Fluency Literal**, mean ± SD	24.1 ± 11.2	41.1 ± 12.1	39.4 ± 9.9 ^1^	<0.001 ***	<0.001 ***	0.540
**Verbal Fluency Categorical**, mean ± SD	12.9 ± 5.1	22.9 ± 6.4	19.5 ± 5.4 ^1^	<0.001 ***	<0.001 ***	0.018 *
**MDRS**, mean ± SD	130.6 ± 8.8	140.7 ± 4.9	141.8 ± 2.5 ^1^	<0.001 ***	<0.001 ***	0.505
**Hopkins B**, mean ± SD	5.7 ± 3.0	9.5 ± 2.4	10.2 ± 1.6 ^1^	<0.001 ***	<0.001 ***	0.306
**HADS Depression**, mean ± SD	6.3 ± 3.2 ^1^		3.0 ± 3.0 ^1^			<0.001 ***
**HADS Irritability**, mean ± SD	7.1 ± 3.3 ^1^		4.6 ± 2.2 ^1^			0.001 **
**HADS Anxiety**, mean ± SD	6.0 ± 3.3 ^1^		4.0 ± 2.6 ^1^			0.016 *
**PBA Depression**, mean ± SD	1.3 ± 2.8 ^1^		0.3 ± 0.8 ^1^			0.049 *
**PBA Irritability**, mean ± SD	0.5 ± 1.4 ^1^		0.3 ± 0.7 ^1^			0.374
**PBA Anxiety**, mean ± SD	3.4 ± 3.5 ^1^		1.2 ± 1.5 ^1^			0.002 **
**PBA Apathy**, mean ± SD	0.5 ± 1.8 ^1^		0.2 ± 0.8 ^1^			0.348
**PBA Total**, mean ± SD	7.0 ± 8.3 ^1^		2.0 ± 2.6 ^1^			0.003
**cUHDRS**, mean ± SD	10.1 ± 3.0	16.9 ± 1.6	17.2 ± 1.3 ^1^	<0.001 ***	<0.001 ***	0.638

^1^ 24 HD and 31 controls from RepairHD. ^2^ Student’s and Chi-squared tests are used to calculate *p*-values: * *p*-value < 0.05, ** *p*-value < 0.01, and *** *p*-value < 0.001. Abbreviations: CAP = CAG age product, cUHDRS = Composite Unified Huntington’s Disease Rating Scale, HD = Huntington’s Disease, PreHD = presymptomatic stage, % = percent of participants, HADS = Hospital Anxiety and Depression Scale, MDRS = Mattis Dementia-Rating Scale, *n* = number of participants, SD = standard deviation, SDMT = Symbol Digit Modalities Test, PBA = Problem Behaviors Assessment Scale, TFC = Total Functional Capacity, and TMS = Total Motor Score.

**Table 2 jpm-12-00174-t002:** Longitudinal decline in HD patients.

Variable	*n*	Number of Visits	Estimated Mean	*p*-Value ^1^
**Follow-up duration in years**, mean ± SD	66	149	1.07 ± 1.62	-
**Number of visits**, mean ± SD	66	149	2.26 ± 1.47	-
**Estimated annual change**, mean ± SE				
TFC	66	144	−0.53 ± 0.06	<0.001 ***
Functional Assessment	66	144	−1.00 ± 0.13	<0.001 ***
Independence Scale	66	144	−3.30 ± 0.36	<0.001 ***
TMS	66	143	2.63 ± 0.46	<0.001 ***
Stroop Word	66	139	−2.71 ± 0.70	<0.001 ***
Stroop Color,	65	138	−1.90 ± 0.72	0.009 **
Stroop Interference	65	138	−1.64 ± 0.49	0.001
SDMT	66	143	−1.47 ± 0.29	<0.001 ***
Verbal Fluency Literal	66	143	−1.11 ± 0.47	0.019 *
Verbal Fluency Categorical	66	143	−0.64 ± 0.23	0.006 **
MDRS	65	140	−0.57 ± 0.30	0.061
Hopkins B	65	142	−0.26 ± 0.14	0.072
HADS Depression	32	88	0.40 ± 0.29	0.182
HADS Irritability	32	88	−0.16 ± 0.28	0.564
HADS Anxiety	32	88	0.71 ± 0.30	0.020 *
PBA Depression	32	88	0.21 ± 0.23	0.369
PBA Irritability	32	88	−3.19 × 10^−4^ ± 0.13	0.998
PBA Anxiety	32	88	−0.10 ± 0.26	0.703
PBA Apathy	32	88	−5.90 × 10^−3^ ± 0.15	0.969
PBA Total	32	60	0.50 ± 0.80	0.536
cUHDRS	66	138	−0.74 ± 0.09	<0.001 ***

^1^ *p*-value refer to the test of significant change over time. Estimated mean annual change, standard error, and *p*-value are estimated with linear mixed model adjusted for age, sex, CAG repeats, and years of education. * *p*-value < 0.05, ** *p*-value < 0.01, and *** *p*-value < 0.001. Abbreviations: cUHDRS = Composite Unified Huntington’s Disease Rating Scale, HADS = Hospital Anxiety and Depression Scale, MDRS = Mattis Dementia-Rating Scale, *n* = number of participants, PBA = Problem Behaviors Assessment Scale, SD = standard deviation, SDMT = Symbol Digit Modalities Test, SE = standard error, TFC = Total Functional Capacity, and TMS = Total Motor Score.

**Table 3 jpm-12-00174-t003:** Items distribution and correlations with domain scores.

Item	Missing (*n* = 66)	Mean ± SD	Pearson’s Correlation		Item Distribution					
			Social Motivation	Emotional Insight	Absolutely True	True	Mostly True	Mostly False	False	Absolutely False
**I am afraid of boring my relatives**	0 (0%)	−0.50 ± 1.63	−0.61	-	4 (6.1%)	12 (18.2%)	12 (18.2%)	4 (6.1%)	20 (30.3%)	14 (21.2%)
**I am interested in others**	1 (1.5%)	1.35 ± 0.91	0.66	-	16 (24.6%)	28 (43.1%)	17 (26.2%)	3 (4.6%)	1 (1.5%)	0 (0%)
**My relatives are not doing enough to help me**	1 (1.5%)	−1.42 ± 1.25	−0.59	-	2 (3.1%)	3 (4.6%)	3 (4.6%)	6 (9.2%)	27 (41.5%)	24 (36.9%)
**I like to make my relatives smile and laugh**	1 (1.5%)	1.15 ± 1.11	0.67	-	14 (21.5%)	27 (41.5%)	16 (24.6%)	3 (4.6%)	5 (7.7%)	0 (0%)
**I make jokes**	0 (0%)	0.47 ± 1.57	0.65	-	12 (18.2%)	17 (25.8%)	14 (21.2%)	9 (13.6%)	8 (12.1%)	6 (9.1%)
**I enliven those around me**	1 (1.5%)	0.01 ± 1.55	0.73	-	5 (7.7%)	18 (27.7%)	12 (18.5%)	6 (9.2%)	18 (27.7%)	6 (9.2%)
**I have laughed uncontrollably at least once during the last month**	1 (1.5%)	0.93 ± 1.54	0.65	-	16 (24.6%)	27 (41.5%)	7 (10.8%)	4 (6.2%)	6 (9.2%)	5 (7.7%)
**I have plans for the coming months**	2 (3.0%)	1.38 ± 1.11	0.51	-	20 (31.2%)	26 (40.6%)	12 (18.8%)	3 (4.7%)	2 (3.1%)	1 (1.6%)
**I want to be treated with sensitivity**	2 (3.0%)	1.69 ± 0.87	0.42	-	27 (42.2%)	26 (40.6%)	7 (10.9%)	4 (6.2%)	0 (0%)	0 (0%)
**I have kept my childhood friends**	1 (1.5%)	0.59 ± 1.70	0.57	-	16 (24.6%)	20 (30.8%)	5 (7.7%)	8 (12.3%)	10 (15.4%)	6 (9.2%)
**I am not interested in other people**	2 (3.0%)	−1.08 ± 1.24	−0.66	-	1 (1.6%)	5 (7.8%)	7 (10.9%)	7 (10.9%)	31 (48.4%)	13 (20.3%)
**I am not interested in doing anything**	2 (3.0%)	−1.09 ± 1.14	−0.67	-	0 (0%)	4 (6.2%)	9 (14.1%)	8 (12.5%)	31 (48.4%)	12 (18.8%)
**I am irritable**	0 (0%)	−0.20 ± 1.45	-	0.67	6 (9.1%)	6 (9.1%)	21 (31.8%)	8 (12.1%)	19 (28.8%)	6 (9.1%)
**My family thinks that I am rude**	0 (0%)	−0.86 ± 1.28	-	0.76	1 (1.5%)	6 (9.1%)	11 (16.7%)	9 (13.6%)	28 (42.4%)	11 (16.7%)
**My family thinks that I am deliberately irritable or angry**	1 (1.5%)	−1.38 ± 1.28	-	0.71	2 (3.1%)	3 (4.6%)	5 (7.7%)	4 (6.2%)	28 (43.1%)	23 (35.4%)
**My relatives blame me for not making enough any effort**	1 (1.5%)	−0.75 ± 1.55	-	0.63	3 (4.6%)	10 (15.4%)	8 (12.3%)	6 (9.2%)	23 (35.4%)	15 (23.1%)
**I am hurt by my relatives’ remarks**	1 (1.5%)	−0.78 ± 1.34	-	0.68	3 (4.6%)	5 (7.7%)	10 (15.4%)	8 (12.3%)	31 (47.7%)	8 (12.3%)
**My relatives are demanding of me**	1 (1.5%)	−0.65 ± 1.37	-	0.70	4 (6.2%)	5 (7.7%)	9 (13.8%)	14 (21.5%)	25 (38.5%)	8 (12.3%)
**I am calm**	2 (3.0%)	0.73 ± 1.40	-	−0.59	10 (15.6%)	25 (39.1%)	11 (17.2%)	10 (15.6%)	4 (6.2%)	4 (6.2%)
**I like it when my opinion is taken into account**	3 (4.5%)	1.48 ± 0.85	-	0.38	15 (23.8%)	37 (58.7%)	8 (12.7%)	1 (1.6%)	2 (3.2%)	0 (0%)
**My relatives are irritable**	2 (3.0%)	−0.84 ± 1.47	-	0.78	3 (4.7%)	7 (10.9%)	7 (10.9%)	10 (15.6%)	22 (34.4%)	15 (23.4%)
**I am demanding of my relatives**	1 (1.5%)	−0.25 ± 1.43	-	0.49	1 (1.5%)	15 (23.1%)	17 (26.2%)	5 (7.7%)	20 (30.8%)	7 (10.8%)
**I want to control what my relatives say about me**	3 (4.5%)	−1.04 ± 1.38	-	0.72	4 (6.3%)	3 (4.8%)	4 (6.3%)	10 (15.9%)	28 (44.4%)	14 (22.2%)
**I do not like my family talking about me**	1 (1.5%)	−0.68 ± 1.39	-	0.56	4 (6.2%)	5 (7.7%)	10 (15.4%)	10 (15.4%)	28 (43.1%)	8 (12.3%)

Abbreviations: SD = standard deviation.

**Table 4 jpm-12-00174-t004:** Associations between subdomain scores at baseline and disease progression.

Variable	Social Motivation and Time Interaction			Emotional Insight and Time Interaction		
	Regression Coefficient ± SE	*p*-Value	Cohen’s f^2^ (CI: 95%)	Regression Coefficient ± SE	*p*-Value	Cohen’s f^2^ (CI: 95%)
**TFC**	−0.002 ± 0.001	<0.001 ***	0.16 (0.03; 0.38)	0.002 ± 0.0005	<0.001 ***	0.15 (0.03; 0.37)
**Functional Assessment**	−0.005 ± 0.001	<0.001 ***	0.19 (0.05; 0.43)	0.003 ± 0.001	0.012 **	0.07 (0.00; 0.24)
**Independence Scale**	−0.010 ± 0.004	0.011 *	0.08 (0.00; 0.26)	0.004 ± 0.003	0.189	0.02 (0.00; 0.13)
**TMS**	0.015 ± 0.005	0.003 **	0.11 (0.01; 0.31)	−0.006 ± 0.004	0.087	0.04 (0.00; 0.17)
**Stroop Word**	−0.035 ± 0.007	<0.001 ***	0.32 (0.11; 0.63)	0.014 ± 0.005	0.009 **	0.08 (0.00; 0.25)
**Stroop Color**	−0.026 ± 0.007	<0.001 ***	0.14 (0.03; 0.34)	0.012 ± 0.005	0.026 *	0.06 (0.00; 0.20)
**Stroop Interference**	−0.020 ± 0.005	<0.001 ***	0.18 (0.04; 0.40)	0.012 ± 0.004	0.001 **	0.13 (0.02; 0.33)
**SDMT**	−0.004 ± 0.003	0.194	0.02 (0.00; 0.14)	0.001 ± 0.002	0.550	<0.01 (0.00; 0.08)
**Verbal Fluency Literal**	−0.008 ± 0.005	0.106	0.03 (0.00; 0.15)	−0.002 ± 0.004	0.628	<0.01 (0.00; 0.07)
**Verbal Fluency Categorical**	−0.005 ± 0.002	0.048 *	0.04 (0.00; 0.17)	−0.001 ± 0.002	0.656	<0.01 (0.00; 0.06)
**MDRS**	−0.003 ± 0.003	0.440	<0.01 (0.00; 0.09)	−0.003 ± 0.002	0.236	0.02 (0.00; 0.13)
**Hopkins B**	−0.001 ± 0.002	0.598	<0.01 (0.00; 0.07)	0.0005 ± 0.001	0.678	<0.01 (0.00; 0.06)
**HADS Depression**	−0.004 ± 0.003	0.163	0.02 (0.00; 0.14)	0.004 ± 0.002	0.106	0.03 (0.00; 0.16)
**HADS Irritability**	−0.001 ± 0.003	0.686	<0.01 (0.00; 0.07)	0.003 ± 0.002	0.097	0.04 (0.00; 0.17)
**HADS Anxiety**	0.002 ± 0.003	0.456	<0.01 (0.00; 0.10)	0.001 ± 0.002	0.632	<0.01 (0.00; 0.07)
**PBA Depression**	−0.001 ± 0.002	0.657	<0.01 (0.00; 0.07)	−0.002 ± 0.002	0.240	0.02 (0.00; 0.12)
**PBA Irritability**	0.0003 ± 0.001	0.812	<0.01 (0.00; 0.05)	−0.001 ± 0.001	0.389	<0.01 (0.00; 0.10)
**PBA Anxiety**	0.005 ± 0.002	0.031 *	0.07 (0.00; 0.24)	−0.004 ± 0.002	0.066	0.05 (0.00; 0.20)
**PBA apathy**	0.002 ± 0.002	0.171	0.04 (0.00; 0.21)	−0.002 ± 0.001	0.165	0.04 (0.00; 0.24)
**PBA Total**	0.005 ± 0.009	0.528	<0.01 (0.00; 0.15)	−0.005 ± 0.006	0.492	0.02 (0.00; 0.18)
**cUHDRS**	−0.005 ± 0.001	<0.001 ***	0.36 (0.13; 0.72)	0.002 ± 0.001	0.001 **	0.15 (0.02; 0.39)

Regression coefficient, standard error, *p*-value, and Cohen’s f^2^ derived from linear mixed models adjusted for age, sex, CAG repeats, and years of education. * *p*-value < 0.05, ** *p*-value < 0.01, and *** *p*-value < 0.001. Abbreviations: CI = confidence interval, cUHDRS = Composite Unified Huntington’s Disease Rating Scale, HADS = Hospital Anxiety and Depression Scale, MDRS = Mattis Dementia-Rating Scale, PBA = Problem Behaviors Assessment Scale, SDMT = Symbol Digit Modalities Test, SE = standard error, TFC = Total Functional Capacity, and TMS = Total Motor Score.

## Data Availability

All data are available upon request from corresponding author.

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
