# Peer review of "Self-Reported Social Relationship Capacities Predict Motor, Functional and Cognitive Decline in Huntington’s Disease"

_jpm, 2022, doi:10.3390/jpm12020174_

Round 1

Reviewer 1 Report

The manuscript entitled Self-reported social relationship capacities predict motor, functional, and cognitive decline in Huntington disease, by van Lemercier et al is a very well designed and performed research. Background, methods, results and discussion are very well exposed. Practically I have not worries or questions about the manuscript.

1.- The authors use mean and standard deviation to describe groups (lines 207-209). Have they check normality?. Please, clarify. In the same paragraph, I’m not convinced to use Chi-squared test to compare pairwise differences between groups. Explain this election.

2.- Table 2. Why SD and SE are used instead of only one of these measures?. Explain the reason or use the same for all the variables.

Author Response

The manuscript entitled Self-reported social relationship capacities predict motor, functional, and cognitive decline in Huntington disease, by van Lemercier et al is a very well designed and performed research. Background, methods, results and discussion are very well exposed. Practically I have not worries or questions about the manuscript.

Response: We appreciate the Reviewer’s concerns and comments that give us the opportunity to clarify the baseline and longitudinal description of the participants. We provided below a point-by-point response to the comments.

Point 1: The authors use mean and standard deviation to describe groups (lines 207-209). Have they check normality?. Please, clarify. In the same paragraph, I’m not convinced to use Chi-squared test to compare pairwise differences between groups. Explain this election.

Response 1: Following the Reviewer’s suggestion, we specified that the normality of the variables was checked visually (see in the manuscript 2.4.2. Description of Groups). While in HD patients we observed a bell-shaped distribution, it was more questionable for other groups. In particular, for TFC, Functional Assessment, Independence Scale where all PreHD and controls participants have the exact same value. Thus, reporting the median and interquartile values would be less informative than the mean and standard deviation. Besides, for the sake of a consistent description across subgroups, we decided to use the same descriptive index in all Table 1.

Concerning the Chi-squared test, it was used only to test if the gender distribution differs between groups (2 by 2). We could have used the exact Fisher test which gives the same conclusions.

Point 2: Table 2. Why SD and SE are used instead of only one of these measures?. Explain the reason or use the same for all the variables.

Response 2: In the HD population, we described the dispersion of the follow-up duration and the number of visits with standard deviation (SD) which is the recommended estimator to represent the variability within the sample (Nagele, Br. J. Anaesth., 2003; http://dx.doi.org/10.1093/bja/aeg087). However, the estimated annual changes derive from linear mixed models and report the decline over a 1-year period at the mean (or a representative values) of covariates. Thus, we reported the standard error (SE) which is used in inferential statistics to give an estimate of how the mean of the sample is related to the mean of the underlying population.

Reviewer 2 Report

In this manuscript, the authors present the SRSQ and describe the development phase and successive validation of this novel questionnaire.

Therefore, the authors found that the SRSQ measures the HD patient’s perceived quality of the social relationship and general life quality.

In my opinion, the present paper is interesting and well written. The methodological and statistical procedure is correct, although they should be explained better, and conclusions are logically linked to the results. I feel that it is a relevant paper for readers.

I do have some concerns:

In the Introduction, a mention could be made on other self-assessment questionnaires that, in previous studies, have examined how patients perceive the quality of social relationships. For example, the authors could be trace parallels with other neurodegenerative diseases.

Moreover, the authors could indicate when the study was conducted.

Finally, English language and style are fine, but there are some minor typing or grammar errors. Please carefully check the whole manuscript.

Author Response

In this manuscript, the authors present the SRSQ and describe the development phase and successive validation of this novel questionnaire.

Therefore, the authors found that the SRSQ measures the HD patient’s perceived quality of the social relationship and general life quality.

In my opinion, the present paper is interesting and well written. The methodological and statistical procedure is correct, although they should be explained better, and conclusions are logically linked to the results. I feel that it is a relevant paper for readers.

Response: We are grateful for the constructive critiques provided by the Reviewer on our manuscript. We have revised the manuscript and made some clarifications in the Statistical Analysis section according to his/her suggestion.

I do have some concerns:

Point 1: In the Introduction, a mention could be made on other self-assessment questionnaires that, in previous studies, have examined how patients perceive the quality of social relationships. For example, the authors could be trace parallels with other neurodegenerative diseases.

Response 1: Few questionnaires have been developed to measure how neurodegenerative diseases impact social relationships and only one has been used for self-assessment. They provide an objective quantification of the patient's social interactions both in society and in own personal environment. However, our questionnaire focuses on HD patients’ subjective feelings about social relationships. To our knowledge, such subjective perspective has not been assessed yet in a similar way in other neurodegenerative diseases.

While the reviewer suggested editing the Introduction, we decided to elaborate on the differences between these questionnaires and ours in the Discussion (section 4.5. Perspectives and Conclusion).

Point 2: Moreover, the authors could indicate when the study was conducted.

Response 2: In order to better understand the course of the study, we have specified the dates of first creation of the questionnaire by the focus group, the online pilot survey, and the validation study.

Point 3: Finally, English language and style are fine, but there are some minor typing or grammar errors. Please carefully check the whole manuscript.

Response 3: We edited the manuscript to correct grammatical and spelling errors.